# Learning Diagrams: A Graphical Language for Compositional Training Regimes

**Mason Lary**[*][†]
University at Buffalo

**Richard Samuelson**[*]
University of Florida

**Alexander Wilentz**
Harvard University

**Alina Zare**
University of Florida

**Matthew Klawonn**
Air Force Research Lab
Information Directorate

**James P. Fairbanks**
University of Florida

## Abstract

Motivated by deep learning regimes with multiple interacting yet distinct model components, we introduce *learning diagrams*, graphical depictions of training setups that capture parameterized learning as data rather than code. A learning diagram compiles to a unique loss function on which component models are trained. The result of training on this loss is a collection of models whose predictions "agree" with one another. We show that a number of popular learning setups such as few-shot multi-task learning, knowledge distillation, and multi-modal learning can be depicted as learning diagrams. We further implement learning diagrams in a library that allows users to build diagrams of PyTorch and Flux.jl models. By implementing some classic machine learning use cases, we demonstrate how learning diagrams allow practitioners to build complicated models as compositions of smaller components, identify relationships between workflows, and manipulate models during or after training. Leveraging a category theoretic framework, we introduce a rigorous semantics for learning diagrams that puts such operations on a firm mathematical foundation.

## 1 Introduction

The deep learning literature is rife with training regimes that exhibit non-trivial interactions between distinct models. Examples include multi-modal architectures with vision and language components (e.g Vinyals et al. (2015), Ramesh et al. (2022)), knowledge distillation schemes Hinton et al. (2015), multi-task learning setups, and on. The practitioner who manages such collections often juggles interacting models that at times are training or frozen, sometimes classifiers and sometimes feature extractors, sometimes trained on a single task and sometimes on many. Motivated by such practical settings we introduce a formalism for building models that treats data sets, models, and their interactions as structured data rather than as code. Our primary contribution, the *learning diagram*, is graphical in nature and has a rigorous mathematical semantics, meaning that learning diagrams can be interpreted unambiguously, manipulated with confidence, and related to one another. Learning diagrams are also compositional, meaning that complex training setups can be built from easier to understand pieces.

At the heart of our formalism is the observation that machine learning problems can often be framed as a search for commuting diagrams. In what is arguably the simplest case, we begin with a collection of patterns $x_i \in \mathbb{R}^m$ and labels $y_i \in \mathbb{R}$. We select an architecture $f : \mathbb{R}^m \times \mathbb{R}^n \to \mathbb{R}$ and learn a parameter $\theta \in \mathbb{R}^n$ such that 1 holds for all pairs $(x_i, y_i) \in \mathbb{R}^m \times \mathbb{R}$.

$$f(x_i, \theta) \approx y_i \tag{1}$$

If we model the patterns, labels, and parameters as functions

---

[*]Equal contributions
[†]Corresponding Authors: `masonlar@buffalo.edu`

$$X : l \to \mathbb{R}^m, \quad Y : l \to \mathbb{R}, \quad \theta : 1 \to \mathbb{R}^n, \tag{2}$$

where $l$ is a finite set that indexes patterns and labels, then the equality in 1 means that the following diagram (approximately) commutes, i.e. that the pair of paths starting at $l$ and ending at $\mathbb{R}$ produce the same real valued output for any choice of element in $l$.

$$\Longleftrightarrow X \times \theta \cdot f = Y \tag{3}$$

Anatomically, Diagram 3 is a graph with nodes that represent *spaces* and edges that represent *maps* between them. The goal of parameterized learning is to find $\theta$ such that the diagram comes as close as possible to commuting. We observe that the diagram itself can contribute to this goal if the common codomain of labels $Y$ and function $f$, a role played by $\mathbb{R}$ in the diagram above, is imbued with a loss function. In that case we can measure the difference between predictions of $f$ and ground truth labels $Y$ and train the component models to minimize said difference. As the loss is minimized, the diagram gets closer to commuting.

The process of generating losses is not limited to diagrams with only one pair of paths that must commute. In fact, we will see that many training regimes are designed to make multiple pairs of paths commute all at once. Further, the implementation of training setups as learning diagrams is not an academic exercise. Rather, doing so allows for the construction and manipulation of training setups in a way that would be more difficult without a structured representation. We believe such a shift from learning pipelines as code to learning pipelines as data is needed to address open problems identified by prior work, such as the need for a "programming interface ... to specify that various adapted models are derived from the same pre-trained model" Bommasani et al. (2021), or "tools that will allow us to build pre-trained models in the same way that we build open-source software" Raffel (2021). Our specific contributions are as follows.

- We formalize the process of producing diagrams of models and data and compiling them to loss functions. We imbue learning diagrams with a rigorous semantics by way of category theoretic machinery.

- We introduce a software library, DiagrammaticLearning.jl, that realizes the theory of learning diagrams to supply convenient operations for building and manipulating training setups, both before models are trained and after, when they may be used as components of other training setups.

- We provide several examples of common training paradigms that can be captured as learning diagrams, and show how their implementation as such affords convenient functionality for common ML tasks.

The plan of the paper is to present these contributions in reverse order. To motivate the implementation we will first show in Section 2 how certain popular training setups can be realized using our approach. To do so, we have selected a small sample of classic machine learning papers and re-implemented them using our framework, along the way recovering the results of the original papers and demonstrating the utility of our diagrammatic formalism for building and manipulating models. In Section 3 we describe the beta version of our library, highlighting available features and the engines that make them work. Finally, in Section 4 we cover the mathematical underpinning of learning diagrams that enables the operations realized in our library. We conclude with an examination of related work and directions for future research.

## 2    Learning Diagram Demonstrations

Our goal in this section is to show that our approach to developing training loops is expedient and feature rich without negatively affecting the final performance of the produced models. We do not

aim to outperform our baselines, but rather to recreate them. In so doing, we will motivate the increasingly abstract constructions of later sections. To facilitate intuition before rigorous definitions, we will leave some terms undefined for now.

We have identified training setups from classic papers that are both representative of widely used paradigms, and are sufficiently complex to highlight aspects of our approach that are practically useful. A particular paper's omission from our experiments does not necessarily indicate that we cannot model it as a learning diagram, nor that doing so would be a fruitless endeavor. Among the diagrams we have worked out but not included are norm-based regularization techniques and autoencoders.

2.1 LEARNING DIAGRAM NOTATION   In each example that follows we define a learning diagram first by considering domains of data corresponding to datasets, their components (such as images and labels), and the representation spaces induced by architectural choices. Each space has an associated Lawvere metric Lawvere (1973), i.e. is a generalization of a metric space relaxing the symmetry and separation axioms, where distances between points may be infinite. For spaces where elements ought to be comparable and generate a loss, we assign one and denote such a space $Y$ as $(Y, \mathcal{L})$, otherwise we assume that distinct points are infinitely far from one another and denote the space $(Y, \infty)$. The edges in a diagram that connect one domain to another depict models and data. We will see in Section 4 that such spaces and maps between them form an object of mathematical significance called a *category*. Our formal theory requires that parameter spaces also be represented explicitly, but for the sake of exposition we will omit or include them in our illustrations as convenient.

With this recipe for specifying learning diagrams in mind, we move on to our first example of learning diagrams in action: Neural Image Captioning Vinyals et al. (2015). It is the oldest of the group of examples and increases the complexity over standard predictions only slightly, but allows us to introduce some useful features of our approach and library.

2.2 IMAGE CAPTIONING   Vinyals et al. (2015) was the first paper to demonstrate an end to end trainable model for image captioning. Named the "Neural Image Captioner" (NIC), it uses a fixed visual encoder to extract features from an image, and a trainable recurrent component to produce a caption. We can depict the model as in Diagram 4.

$$(N, \infty) \xrightarrow{\langle CNN, Y \rangle} (Z_I \times Y, \infty) \xrightarrow{LSTM} (Z_L, \infty) \qquad (4)$$

$$\searrow^{Label} \qquad \downarrow^{Prediction}$$

$$(Y, \mathcal{L}_{ce})$$

In replicating the NIC diagrammatically, we aim to show that diagrams enable easy swapping of architectures for different components, and easy manipulation of model properties such as whether a model updates or is frozen. The components of the NIC diagram include a graph that captures the "syntax" of our training set-up, as well as "semantic" data associated with the graph that includes the particular choice of architectures and spaces. Such separation can be exploited; whereas the original paper has a single choice of CNN encoder Szegedy et al. (2015), we can easily vary the encoder backbone by simply assigning a different architecture to the CNN edge of the diagram. In other words, there is no need to manually chain together the CNN and LSTM, as our backend handles it for us. The code required for such a switch is depicted in Listing 1, and is only slightly edited from our implementation. We trained three semantic variations of Diagram 4 corresponding to three choices of CNN encoder; GoogLeNet Szegedy et al. (2015), Resnet-50 He et al. (2016), and an image transformer Dosovitskiy (2020). Vinyals et al. (2015) used GoogLeNet in the original paper. By using more powerful architectures and replicating other portions of the model and training procedures, we were able to outperform the original paper's BLEU score Papineni et al. (2002) on the Flickr8k Hodosh et al. (2013) and Flickr30k Plummer et al. (2015) image captioning data sets. The results of our experiments are captured in Table 1.

The mix of trainable and fixed models offers a chance to highlight some model manipulation. In particular, we can utilize model homomorphisms to identify sub-components and apply functions to them. To specify a homomorphism, we first specify the shape of a sub-diagram, then describe a mapping from the sub-diagram to the larger model with components we want to manipulate. In

| Data Set | GoogLeNet | Resnet-50 | ViT-B-16 | Original |
|---|---|---|---|---|
| Flickr8k BLEU | 66.85 | 66.93 | **71.39** | 63 |
| Flickr30k BLEU | 70.06 | **76.72** | 71.78 | 66 |

Table 1: Results of image captioning models trained as learning diagrams, named after the convolutional component.

Listing 1 we show code that allows us to set the vision encoder to "eval" mode. While this is easy enough to do with standard PyTorch for such a simple model, for more complex diagrams with more models it becomes especially advantageous to leverage homomorphisms which can pick out subgraphs based on some structural pattern matching and apply different transformations to the different pieces of the pattern Ehrig et al. (1973).

2.3 KNOWLEDGE DISTILLATION    Much like the simple Diagram 3, the image captioning use case contains only one pair of paths that induce a loss; we want the captions produced by the CNN+LSTM team to be the same as the ground truth. Learning diagrams are capable of handling much more complicated setups where multiple pairs of parallel paths contribute terms to the loss. To demonstrate this capability, we now describe how to implement knowledge distillation as a learning diagram.

Knowledge distillation can refer to many different techniques (Buciluǎ et al. (2006); Hinton et al. (2015)) that share the common goal of taking information from a powerful model or ensemble called the "teacher" and recovering that information in a simpler, easier to use model called the "student." A particularly popular approach to distilling knowledge is to train the student on the output of the teacher, whether such output be logits or a softmax with a temperature. Further, the student is trained on labeled data that can be the same as that used in training the teacher, and so receives supervisory signals from two sources; the outputs of the teacher model and labeled data.

Figure 1 recovers a knowledge distillation setup; notice there are two pairs of parallel paths. One terminates at $Y$, the label space, and the other in $G$, the soft target space. The learning diagram therefore induces a loss with two terms, one corresponding to the predictions of the student model relative to the ground truth labels, and the other corresponding to the softmax outputs of the student model relative to the soft outputs of the teacher. The official PyTorch knowledge distillation introduction  Chariton (2024) has a more detailed specification of the experiments than the original paper Hinton et al. (2015), so we compare to those benchmarks. We see in Table 2 that the learning diagram produces results comparable to those of the official implementation.

```
v_model = load_CNN(cnn_name)
train_ds = get_dataset()
...
diagram = @LD begin
    N ⇒ [train_ds]
    Z_I_x_Y ⇒ (Any, Vector{Float64})
    Z_L ⇒ Any
    Y ⇒ {Vector{Float64}, cross_entropy}
    CNN_x_Y: N → Z_I_x_Y ⇒ x -> (v_model(x[0]), x[1])
    LSTM: Z_I_x_Y → Z_L ⇒ (x, y) -> lstm(x)
    Label: Z_I_x_Y → Y ⇒ (x, y) -> y
    Prediction: Z_L → Y ⇒ x -> predict(x)
    CNN_x_Y · LSTM · Prediction ≤ CNN_x_Y · Label
end
```

Listing 1: Key functionality for the NIC implementation. Users specify diagrams as Python or Julia data structures (Julia shown) similar to an edge list representation of the underlying graph and assign models to components of the diagram. Because the parts of the model are represented as data instead of code, users can specify and manipulate sub-diagrams and components programmatically, without the complexity of manipulating unstructured code. We also demonstrate that mapping to data spaces provides type annotations at different levels of granularity for models in the diagram.

$$(N, \infty) \xrightarrow{\pi_1} (X, \infty) \xrightarrow{s} (S, \infty) \xrightarrow{\sigma} (G, \mathcal{L}_{st})$$

$$\pi_1 \cdot s \cdot \sigma = \pi_1 \cdot t \cdot \sigma$$
$$\pi_1 \cdot s \cdot \sigma = \pi_2$$

Figure 1: A diagrammatic representation of Knowledge Distillation (left), the red and blue parallel paths encode equations that the optimization procedure attempts to solve (right). Components models used in both pairs of parallel paths are shown in violet.

|                          | Teacher | Student | Distilled Student |
|--------------------------|---------|---------|-------------------|
| PyTorch Implementation   | 74.72   | 70.50   | 70.81             |
| Learning Diagram         | 76.24   | 67.67   | 69.21             |

Table 2: Performance comparison of official PyTorch knowledge distillation vs DiagrammaticLearning.jl on CIFAR-10.

2.4  FEW SHOT LEARNING   We now turn to an example that will highlight the utility of building more complicated models in a compositional way. The multi-task and few-shot learning literature contains many examples that would benefit from a compositional perspective, thanks to their pervasive use of shared backbones in conjunction with task specific modules. We select Tian et al. (2020) as our replication target to demonstrate the utility of a compositional perspective, since it exemplifies the idea of training a feature extractor on as much available data as one can, and subsequently training small task specific modules that exploit the learned features. More convoluted training pipelines have had limited success in improving upon this approach. In the case of Tian et al. (2020), the data used comes from Tiered Imagenet Ren et al. (2018), Mini-Imagenet Vinyals et al. (2016), FC100 Oreshkin et al. (2018), and CifarFS Bertinetto et al. (2018). A ResNet-12 backbone is trained on all data, and subsequently few shot tasks are sampled at random from the respective test sets. Simple affine classifiers are then trained on the feature-label pairs extracted by the pre-trained backbone.

The meta-train and meta-test steps of Tian et al. (2020) have very similar syntax, in that they both extract features and train a classifier to make predictions using those features. The difference is that during meta-training the feature extractor is updated on all tasks. Intuitively, if we can track the task specific interactions of data and models and use them to define the composite interaction seen during meta-training, then we can easily re-use them during meta-testing and avoid implementing a new meta-testing loop from scratch. Learning diagrams provide such functionality by allowing us to define task specific classification setups, and subsequently glue those setups along the shared axis of a common encoder. The exact construction we use for composition, a *colimit*, is depicted and described in Figure 2.

| Approach                   | MI 1-Shot  | MI 5-Shot  | TI 1-Shot    | TI 5-Shot    |
|----------------------------|------------|------------|--------------|--------------|
| Tian et al. (2020) Simple  | 62.02      | 79.64      | 69.74        | 84.41        |
| Learning Diagram           | 58.13      | 78.82      | 66.48        | 80.15        |
| Approach                   | CFS 1-Shot | CFS 5-Shot | FC100 1-Shot | FC100 5-Shot |
| Tian et al. (2020) Simple  | 71.5       | 86.0       | 42.6         | 59.1         |
| Learning Diagram           | 70.5       | 85.6       | 43.1         | 59.3         |

Table 3: Performance comparison of Tian et al. (2020) vs DiagrammaticLearning.jl on Mini-Imagenet (MI), Tiered-Imagenet (TI), Cifar-FS (CFS), and FC100.

The compositional perspective not only allows us to build large training pipelines quickly, but also to track the presence of notable sub-diagrams via the inclusion morphisms computed as part of the colimit computation. As a result, upon training the foundation model, we can efficiently proceed to the meta-test phase. Using the same "edata" syntax as in Listing 1, we first assign a new classifier head and N-way K-shot data set to each of the classifier sub-diagrams, set the encoder to "eval

$$
\text{colim} \left( \begin{array}{c} N \xrightarrow{\ x\ } X \xleftarrow{\ \ \ \ \ } \bullet \dashrightarrow X \\ {\scriptstyle y}\downarrow \qquad\qquad\qquad\qquad\qquad \downarrow{\scriptstyle m} \\ Y \xleftarrow{\ \ \ \ \ } \bullet \dashrightarrow Y \xleftarrow{\ h\ } F \end{array} \right) = \begin{array}{c} N \xrightarrow{\ x\ } X \\ {\scriptstyle y}\downarrow \quad \downarrow{\scriptstyle m} \\ Y \xleftarrow{\ h\ } F \end{array}
$$

$$
\text{colim} \left( \begin{array}{c} N_1 \qquad\qquad\qquad N_2 \\ {\scriptstyle y_1}\downarrow \ {\scriptstyle x_1}\searrow \quad\qquad {\scriptstyle x_2}\swarrow \ \downarrow{\scriptstyle y_2} \\ \quad X \dashleftarrow \bullet \dashrightarrow X \\ Y_1 \quad {\scriptstyle m}\downarrow \qquad\quad \downarrow{\scriptstyle m} \quad Y_2 \\ {\scriptstyle h_1}\nwarrow \ F \dashleftarrow \bullet \dashrightarrow F \ {\scriptstyle h_2}\nearrow \end{array} \right) = \begin{array}{c} N_1 \qquad\qquad N_2 \\ {\scriptstyle y_1}\downarrow \ {\scriptstyle x_1}\searrow \ {\scriptstyle x_2}\swarrow \ \downarrow{\scriptstyle y_2} \\ \quad X \\ Y_1 \ {\scriptstyle m}\downarrow \ Y_2 \\ {\scriptstyle h_1}\nwarrow F \nearrow{\scriptstyle h_2} \end{array}
$$

Figure 2: Learning diagrams can be built using the categorical construction of a colimit, which generalizes the concept of a set union or quotienting by an equivalence relation. **(Top)** A data set and classifier/encoder pair are composed by identifying the common image and label domains $X$ and $Y$ (dashed arrows). The result of the colimit computation is a square whose induced loss trains the classifier and encoder on the labeling task of the associated data set. **(Bottom)** Two classifier squares are merged along a common feature extraction backbones $m$. The resulting colimit is a learning diagram whose loss trains classifier heads on their respective tasks and the common backbone model on all tasks. For this colimit to be well specified, both the image space $X$ and the feature space $F$ have to be identical. Additional classification heads can be attached by further colimits.

mode", build the model, and then train on a fewshot data set. Repeating this procedure per randomly sampled few-shot data set gives us the performance noted in Table 3.

## 3 IMPLEMENTATION

Having motivated the functionality of DiagrammaticLearning.jl * with some common ML tasks, we now turn to describing the implementation that provides such functionality.

### 3.1 SPECIFYING THE DIAGRAM DATA
To generate a composite loss, users must specify a diagram. Listing 1 shows an example of the structures required by the library. The required components are

- Vertices with their labels, a mapping to a data space, and an optional distance metric. Data spaces are captured by Julia types, which allows the user to annotate their diagram with type information for computations. This can be circumvented by supplying `Any` as a type within the diagram.
- Edges with their labels, their source and target vertices, and a mapping to the computation the edge performs. These computations are simply Python or Julia functions.
- Relationships between edges, represented with $\leq$. This requirement specifies a preorder on the edges of a graph, a concept specified further in Section 4. It can also be seen as a specification of which edges in the graph should take part in loss calculations.

### 3.2 FINDING PARALLEL PATHS
From a learning diagram, we aim to calculate a single loss function. Since the loss function is a sum over all the parallel paths, we first need to compute all pairs of parallel paths from the graph.

To accomplish this, we apply an algebraic perspective and use the fact that powers of the adjacency matrix count paths in a graph. This approach is a generalization of the celebrated Floyd-Warshal algorithm Floyd (1962). We represent the graph as a $V \times V$ matrix with entries that are sets of edges

---

*https://github.com/AlgebraicJulia/DiagrammaticLearning.jl

in the graph. This allows us to define a generalized version of matrix multiplication which computes paths between vertices Fong and Spivak (2018); Pratt (1989). Multiplication of elements represents concatenation of paths and summation represents union of sets, which leads to an algorithm that computes all the paths in a graph. Powers of this set-of-paths valued matrix then enumerate the paths between each pair of vertices of a fixed length. Parallel paths can then be collected from any matrix element with more than 1 element.

Many common machine learning loss functions are not symmetric. These loss functions require parallel paths to occur in a specified order. To facilitate this, we assume that all pairs of single edges are ordered (i.e. $e_1 \leq e_2$ and $e_2 \leq e_1$ for edges $e_1, e_2$), unless only a single pair is specified in the graph. The process that builds the matrix containing parallel paths additionally builds up a preorder between paths, allowing a filter to include only parallel paths that conform to the preorder.

3.3 INTEGRATION WITH EXISTING LIBRARIES    Due to the generic nature of our representation, we can view a learning diagram as a sort of intermediate language, independent of a particular neural network framework. To this end, we can compile either PyTorch or Flux models from a given diagram. This flexibility allows for the creation of diagrams for new workflows and experiments, as well as the retrofitting of diagrams to existing codebases. This paper focuses on PyTorch models due to its ubiquity in machine learning research and practice.

The correctness of the compilation process is explored rigorously in the next section, but for a more familiar explanation, we refer to PyTorch components. Vertices in a graph in our framework contain both data sources (such as a PyTorch dataloader) as well as a loss function. A path in our graph can quite easily be converted into a PyTorch Sequential model. Given parallel paths, a module can be created which samples $x$ from the data source present at the domain of the paths, passes $x$ through both sequential models to obtain outputs $y, \hat{y}$, and outputs $l(y, \hat{y})$, the result of applying the loss function $l$ found in the codomain of the paths. Summation of these modules represents composite loss functions. This process thus encapsulates data sampling, network computation, and loss computation into a single PyTorch module.

The requirements for compatibility with our framework are similar to those for compatibility with pure PyTorch. For data, we require iterable components, whether it be DataLoaders or just lists. For computations, we require functions, either in the form of torch modules, or pure Python functions. Thus, popular frameworks within the PyTorch landscape, such as models from HuggingFace, are compatible with our system.

## 4    THEORETICAL GROUNDING

Having discussed some applications and our implementation, we now turn to the underlying category theoretic formalisms and definitions. It is the content of this section that ensures learning diagrams have rigorous semantics, and therefore that manipulations of diagrams and maps between them can be interpreted mathematically. The ultimate goal of the section is a machinery that "mathematically compiles" a learning diagram into a composite loss function that drives participating models to form approximately commuting diagrams. Throughout this section we will use some category theory that we may not define due to space constraints. Everything left undefined should be easy to find in introductory references Riehl (2017); Fong and Spivak (2018). Our story proceeds by defining what a learning diagram is, explaining how parallel paths can be extracted from it, and finally how these parallel paths combine to form a composite loss function.

4.1    LEARNING GRAPHS AND LEARNING DIAGRAMS    A *learning graph* $(G, \leq)$ is a directed multigraph $G = (V, E, \text{src}, \text{tgt})$ together with a preordering $\leq$ of its edges.

We will motivate the preorder in due time. On its own, a learning graph doesn't associate a loss function to each node of the graph. Rather, we consider it a syntactic presentation of a training regime. As we saw in the image captioning example, this distinction can be useful when varying the exact data or models being used. In the end, however, what we want is a full learning diagram. The choice of the word diagram is no accident; it has a proper category theoretic definition.

**Definition 4.1.** (1.6.4 of Riehl (2017)) A diagram in a category $\mathsf{C}$ is a functor $D \colon \mathsf{J} \to \mathsf{C}$ whose domain, the indexing category, is a small category.

To define a diagram, therefore, we must first specify a domain category J and a codomain category C. Our domain category comes from our learning graph and identifies the syntactic structure of our training regime: how many spaces of data or representations we are considering, how many models, etc. The codomain category C defines what kind of thing each node and edge in our learning graph is, i.e. the semantics of the training setup, and the functor $D$ is the mapping from syntactic structure to semantic content that assigns particular spaces and models to edges. Constructing a category to serve as our domain is fairly straightforward; we can take a learning graph and turn it into a category by taking vertices of the graph to be objects of a category and *paths* in a graph to be morphisms. Such a construction is called the *free category on a graph*. Defining the semantics conferring codomain requires more slightly more work. Given that we want to associate a potentially asymmetric loss function to each node in a learning graph, we should consider a category C whose objects are some generalization of metric spaces. We choose the category $Law$.

**Definition 4.2.** Denote by Law the category that has objects Lawvere Metric Spaces Lawvere (1973) and morphisms Lipschitz continuous functions.

4.2 FINDING PATHS AND BUILDING LOSSES  Let Par be the free category on the following graph.

$$\bullet \quad \overset{+}{\underset{-}{\rightleftarrows}} \quad \bullet \tag{5}$$

We can use the category Par (5) to formalize the construction of losses from parallel paths. If $(G, \leq)$ is a learning graph, and if $\mathsf{Free}(G)$ is the free category generated by $G$, then a functor $P : \mathsf{Par} \to \mathsf{Free}(G)$ chooses a pair of paths in $G$ with the same initial and terminal vertex. If $D : \mathsf{Free}(G) \to \mathsf{Law}$ is a learning diagram on $G$, then the composite functor $D \circ P : \mathsf{Par} \to \mathsf{Law}$ chooses a pair of Lipschitz continuous functions $f : X \to Y$ and $g : X \to Y$ with the same domain and codomain. We will assign a loss to these functions as follows. If $X$ is a finite set, then we define $\ell(f, g) \in [0, +\infty]$ to be the sum

$$\ell(f, g) := \sum_{x \in X} d_Y(f(x), g(x)).$$

If $X$ is an infinite set, then we take the supremum over all finite sums. This quantity measures the distance between outputs of $f$ and $g$ summed over all inputs. It also has the following contractive property: if $f : X \to Y$, $g : X \to Y$, $f' : X' \to Y'$, and $g' : X' \to Y'$ are Lipschitz continuous functions, and if $p : X \to X'$ and $q : Y \to Y'$ are surjective 1-Lipschitz functions such that

$$f'(p(x)) = q(f(x)) \text{ and } g'(p(x))) = q(g(x)))$$

for all $x \in X$, then $\ell(f, g) \geq \ell(f', g')$. Formally, loss is an enriched functor $\ell : \mathsf{C} \to \mathsf{Cost}$, where C is the subcategory of the functor category $\mathsf{Law}^{\mathsf{Par}}$ whose fibers are surjective 1-Lipschitz functions, and where Cost is the partially ordered set $[0, \infty]$.

**Theorem 4.3.** *The mapping $P \mapsto \ell(f, g)$ defines a functor $\ell : \mathsf{C} \to \mathsf{Cost}$.*

*Proof.* Let $f$, $g$, $f'$, $g'$, $p$ and $q$ be the functions described above. Functoriality is the contractive property, $\ell(f, g) \geq \ell(f', g')$. To see this, let $S \subseteq X'$ be a finite set. Then,

$$\sum_{x' \in S} d_{Y'}(f'(x'), g'(y')) \leq \sum_{x \in p^{-1}S} d_{Y'}(f'(p(x)), g'(p(x)))$$

$$= \sum_{x \in p^{-1}S} d_{Y'}(q(f(x)), q(g(x)))$$

$$\leq \sum_{x \in p^{-1}S} d_Y(f(x), g(x))$$

where the first line follows from the surjectivity of $p$ and the third from the contractivity of $q$. The desired inequality follows from the arbitrary selection of $S$. $\square$

4.3  LIMITING THE SEARCH  Diagrams in Law with a shape given by a learning graph very nearly capture the data we require to build a composite loss, but for two problems. For one, if we are allowing asymmetric loss functions, then we must provide the creator of a learning diagram with a way to say how arguments to a loss should be ordered. Additionally, some vertices in the learning graph will be assigned finite collections for indexing data sets and other vertices will be assigned vector spaces for representing the spaces containing the data values. The loss functions should only use parallel paths that start at indexing, because these lead to finite sums in the loss functions.

**Ordering paths**  The preorder on edges in a learning graph naturally leads to a preorder structure on the free category associated with a graph. This produces a *locally posetal 2-category*, which is a category with a preorder on every hom-set, such that morphism composition is monotonic.

The upshot is that our syntactic 2-category now tracks user specified ordering of models, i.e. which model's predictions should be the first argument to a loss and which should be the second.

**Indexing and Nonindexing Vertices**  The other problem is that we only want to build composite losses for parallel paths that start at specific nodes. The machinery of categories provides a succicint way to specify the relevant constraints. Let $\mathsf{Ind}$ be the free 2-category generated by the learning graph

$$i \longrightarrow n \tag{6}$$

. By setting $- \leq +$, we may upgrade the category $\mathsf{Par}$ defined in section 2 to a 2-category. These categories are related by a 2-functor $\alpha : \mathsf{Par} \to \mathsf{Ind}$.

$$\tag{7}$$

We use the 2-category $\mathsf{Ind}$ to assign labels to each vertex of a learning graph. Specifically, for all learning graphs $G$, a functor $\beta : \mathsf{Free}(G) \to \mathsf{Ind}$ is a labelling of the vertices of $G$ as either *indexing* or *non-indexing*, and the structure of $\mathsf{Ind}$ ensures that indexed vertices precede non-indexed ones. A commutative diagram of the form (8) choses a pair of parallel paths that begin at indexed nodes.

$$\mathsf{Par} \xrightarrow{P} \mathsf{Free}(G)$$
$$\alpha \searrow \quad \downarrow \beta \tag{8}$$
$$\mathsf{Ind}$$

Given a learning diagram $D : \mathsf{Free}(G) \to \mathsf{Law}$, the composite functor $D \circ P : \mathsf{Par} \to \mathsf{Law}$ chooses a pair $f : X \to Y$ and $g : X \to Y$ of Lipschitz continuous functions, and we can compute the loss of these functions as in section 2. (We define the *composite loss* of a triple $(G, D, \beta)$ to be the sum $\ell^*(G, D, \beta) \in [0, +\infty]$ given by

$$\ell^*(G, D, \beta) = \sum_p \ell(f, g)$$

) where $f$ and $g$ are the functions corresponding to the functor $D \circ P$, and where the sum is taken over all choices of $P$ making the the diagram in 8 commute. This construction allows us to build losses automatically from graphical specifications of machine learning architectures. The contractive nature of loss functions ensures that composite losses are well behaved with respected to taking subsets of each dataset in the diagram.

## 5  RELATED WORK

We find ourselves aligned in vision with the field of AutoML in that we seek to make the life of the machine learning practitioner easier. More specifically, we fit with work that focuses on structuring machine learning pipelines; a pipeline focused AutoML survey can be found in Zöller and Huber (2021). The formulation of Zöller and Huber (2021) describes pipelines as Directed Acyclic Graphs

(DAGs). We differ from this formulation and other DAG-based formulations in many important ways. For one, we have a mathematically grounded definition of what data is captured by nodes and paths in our learning diagrams, whereas prior art appeals to imprecise terms such as "primitives" Lippmann et al. (2016), "algorithms", "operators", "transformations" and so on. There are some works (e.g. Drori et al. (2019)) that place more structure on pipelines, but in this framework the grammar is still defined on an unstructured set of ad-hoc operations. Further, works Kalyuzhnaya et al. (2020); Nikitin et al. (2022); Polonskaia et al. (2021) that define somewhat flexible pipelines usually limit themselves to DAGs that have a single source and sink representing the data and label space. As best we can tell, there appear to be a few exceptions to this rule, e.g. Klein et al. (2017) which is limited to Bayesian settings with a common data domain shared among tasks. Thanks to the rigor of our model we can accommodate arbitrarily many tasks interacting in arbitrary ways, so long as the nodes are Lawvere metric spaces and the paths between nodes are continuous.

Learning diagrams also provide a graphical programming language for constructing machine learning models, and in that sense are related to works that study foundations of languages for machine learning models Elliott (2018), though such techniques tend to focus on probabilistic programming Gordon et al. (2014) or the semantics of gradient descent and less on the semantics of the objective to be optimized.

Category theory has already been used to examine machine learning models compositionally, for example in Fong et al. (2019); Cruttwell et al. (2022); Gavranović (2022). Our approach is somewhat different from most such works in that we focus on the semantics of the loss or objective function as opposed to the semantics of model updates. That said, we believe there are connections to explore, particularly with Hanks et al. (2024) as it also considers how to specify compositionally a loss for machine learning models. More generally, our work belongs to an emerging tradition of using category theory to provide a diagrammatic yet rigorous approach to describing models in a domain of interest Decapodes.jl Morris et al. (2024); Patterson et al. (2023); Libkind et al. (2022) and AlgebraicPetri.jl Libkind et al. (2022).

## 6 Future Work and Conclusion

We introduce DiagrammaticLearning.jl, a library built on the notion of learning graphs and learning diagrams allowing machine learning practitioners to specify complex interactions between multiple models in a way that is inherently compositional and structured. The utility of this approach is established by capturing some popular training setups as learning diagrams and demonstrating convenient functionality that came as a result. We further outline the implementation and theoretical backing that make learning diagrams possible, demonstrating a rich and rigorous semantics that leads naturally to principled operations for manipulating structured collections of models.

This paper has only scratched the surface of potential capabilities. Future work should investigate how model structure can be exploited to improve distributed training and manage high performance systems. Additionally, our structured representation seems like a candidate for the foundation of a more complete machine learning model management system. We hope for example to elevate software centric definitions of reproducibility, such as the one given in He et al. (2021), to a mathematical plane that is "implementation free" in the sense that any implementation faithfully reproducing the math will yield equivalent results. Homomorphisms between diagrams could capture relationships between development iterations and facilitate operations like searching for models with particular properties or component sub-diagrams. On the more theoretical side, our formalism is currently limited to building losses for finite data sets, but extensions to methods where continuous distributions can enter the diagrams are possible. Additionally, our implementation of training networks of models uses mini-batches, but the theory does not address such aspects of practical training algorithms.

Development and adoption of these tools would reduce the labor necessary to produce effective ML pipelines, accelerate iteration of ideas within and between research groups, reduce errors introduced in software implementations, and provide quicker turnarounds while evaluating novel ideas. We hope the learning diagrams paradigm will accelerate the overall process of machine learning research.

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
