# OpenReview forum: "Learning Diagrams: A Graphical Language for Compositional Training Regimes"
_ICLR.cc/2025/Conference — ICLR 2025 Poster_

### Official Review · Reviewer_KsYd · 2024-11-01

**Soundness:** 3
**Presentation:** 2
**Contribution:** 2
**Rating:** 6
**Confidence:** 3

**Summary:**

The paper addresses the problem of formally describing training protocols. For this purpose, they introduce a graphical formalisation called *learning diagrams*. First, they motivate their proposal by providing a series of examples of how common training paradigms can be accurately and concisely described via learning diagrams (namely image captioning, knowledge distillation, and few-shot learning). Then, they introduce a library called DiagrammaticLearning, that provides an interface for describing and building training setups. Finally, they formalise the process of describing models and data and compiling them into loss functions using category theory.

The library, called *DiagrammaticLearning.jl*, provides an interface to describe learning diagrams as labeled directed graph. Intuitively, vertices describe data sources and losses, and edges represent models, which in turn are functions applied to the data. The library then detects and sums over parallel paths to determine a single loss function. Finally, the library can convert the learning diagram into an implementation using a deep learning framework (e.g. PyTorch). Thus, the library is an intermediate language to describe training protocols and converting them into actual implementations.

**Strengths:**

- Communicating training setups used is often a significant challenge and can lead to miscommunication, especially in scenarios where multiple models interact with each other. Formally specifying training setups provides an opportunity to address this problem.
- The series of continuously more complex examples makes it easy to follow the ideas behind learning diagrams.
- The authors evaluate their framework against the original implementations of the selected examples and demonstrate that it achieves similar performance.

**Weaknesses:**

- Despite using an intuitive graphical representation, the code in listing 1 is hard to read and not particularly intuitive.
- Section 3 contains little details about the actual implementation.
- While the authors mention that they have removed the code for the double blind review process, I would have appreciated anonymised code in the supplementary material considering that the framework is the main contribution. This would have allowed the reviewers to interact with the framework, provide more detailed feedback regarding functionality and usability, and see whether it actually supports development in the ways described in the paper.

**Questions:**

- Could you elaborate on the role of DiagrammaticLearning as a compiler (Section 3.3)? How exactly does it compile the graph into e.g. PyTorch training code?
- Is the framework compatible with common PyTorch frameworks, e.g. HuggingFace?

---

> ### Author Response · Authors · 2024-11-23
>
> Thank you for the review! We appreciate the chance to find areas of improvement in the less abstract portions of our work, especially as we aim for this to be a helpful tool for machine learning practitioners.
>
> > Could you elaborate on the role of DiagrammaticLearning as a compiler (Section 3.3)? How exactly does it compile the graph into e.g. PyTorch training code?
>
> We agree that our implementation details were too light. We have updated Section 3.3 to address these concerns and provide details in PyTorch specific terms. We hope this provides more clarity into our compilation process.
>
> > Is the framework compatible with common PyTorch frameworks, e.g. HuggingFace?
>
> Yes, we discuss the generality of our approach in Section 3.3, albeit briefly. We've updated the language of this section in order to make it clear the variety of frameworks that can fit into our formalization. In essence, if other tools are compatible with standard PyTorch network components and losses, they are compatible with our framework.
>
> > Despite using an intuitive graphical representation, the code in listing 1 is hard to read and not particularly intuitive.
>
> This is ongoing future work to develop an DSL on top of this framework to make this easier. In the interim, our framework supports building these learning diagrams in Julia, where we can utilize several language features to write what we think is clearer code. We've updated our code listing to this Julia syntax.

---

> > ### Comment · Reviewer_KsYd · 2024-11-27
> >
> > Thank you for your reply. I have read the other reviews and the authors' responses.
> >
> > I appreciate the rigorous framework for describing training systems and believe that it could help to improve building, maintaining, and reproducing machine learning training approaches. My main concern remains usability - at this stage, I am not convinced that the framework is in a state to facilitate widespread adoption. I believe that a DSL built on this framework could help to achieve this. However, I also believe that ICLR should expose researchers and practitioners to new ideas and techniques that may advance their research and practice. Therefore, I raise my score to 6.

---

### Official Review · Reviewer_VJye · 2024-11-01

**Soundness:** 3
**Presentation:** 4
**Contribution:** 3
**Rating:** 8
**Confidence:** 4

**Summary:**

The paper proposes a new approach to managing complex ML training setups called learning diagrams. It offers a new, more structured, way of looking at compositionality and interaction between different components (like data and different models) in composite learning settings. More specifically, in the learning diagrams framework nodes represent objects within a training regime, which can be data spaces, intermediate representations or parameter spaces. Each node, representing these spaces, is also associated with a distance measure -- a loss function. The edges represent transformations between two spaces (can be a learnable function represented with a neural net). Learning amounts to finding parallel passes that connect the same start and end nodes, and using their composite loss for learning. Composite loss is the sum of all pairwise losses of the parallel paths through a learning diagram.

The proposed concept of learning diagrams is rigorously grounded in the category theory. Additionally, the paper introduces a software framework that enables the specification of learning diagrams as a Python data structure.

**Strengths:**

**Originality**:
The ideas presented in this work appear to be original in several ways. The paper applies category theory to create a formal framework that generalizes to different common training setups in ML. It proposes a way to automatically find parallel passes that enable the calculation of the composite loss used for training. Additionally, the paper proposes to view a training setup as a manipulable graphical object with adjustable and replaceable components (as data rather than code). I think this idea is original and leads to a nice generalization of many existing training regimes potentially opening doors for new interesting training setups.

**Quality&Clarity**: The paper appeared very well structured to me. The clarity of exposition is generally very good.

**Significance**: the significance of the work lies in providing a theoretical and software framework for designing multi-component learning systems. It facilitates compositionally in ML system design, which can lead to more maintainable and efficient model development (subparts can be easily replaced with new ones). It also facilitates collaborative model development through the integration of a plethora of independently trained models available online. The idea of representing training frameworks as a data structure (rather than code) can be impactful by increasing the transparency and interpretability of different complex training setups. It can help gain insights into relationships between different training regimes.

**Weaknesses:**

Learning diagrams provide a unified framework that encompasses many different setups, and this is demonstrated in this work by representing several training settings as in the language of learning diagrams.

However, I think this paper could benefit from some concrete demonstrations of the advantages of learning diagrams over conventional methods in terms of performance, interpretability, training efficiency, revealing unknown connections between existing training setups etc..

**Questions:**

- can the framework presented here be used to enable a type of modularity where at each intermediate "layer" of a modular learner a routing decision has to be made (similar to MoEs/MoErging methods https://www.arxiv.org/pdf/2408.07057 )? i.e. here intermediate modules do not necessarily produce outputs that are in the label space of the underlying data

---

> ### Author Response · Authors · 2024-11-23
>
> Thank you for your review, we share your excitement about the possibilities of learning diagrams and agree that we have yet to demonstrate all the potential advantages that the formalism could confer. We aim to do so in future work!
>
> As for applicability to model MoErging, we believe that we can certainly handle set ups where expert aggregation is done on the "output" as defined in https://www.arxiv.org/pdf/2408.07057. The generic pattern shared between such approaches appears to be like that depicted at the following link, where a number of experts $f_1$, $f_2$, $f_3$ operate on "copies" of an input datum and a fusion module makes predictions, as seen in [this diagram](https://q.uiver.app/#q=WzAsNCxbMCwwLCJOIl0sWzIsMCwiWF4zIl0sWzIsMiwiRiJdLFswLDIsIlkiXSxbMCwxLCIoeCx4LHgpIiwxXSxbMCwzLCJ5IiwxXSxbMiwzLCJmdXNpb24iLDFdLFsxLDIsImZfMSIsMSx7ImN1cnZlIjoyfV0sWzEsMiwiZl8zIiwxLHsiY3VydmUiOi0yfV0sWzEsMiwiZl8yIiwxXV0=)
>
> We note however that in this diagram, the experts must be held fixed, otherwise optimizing the loss terms induced by the commuting paths would update the experts in addition to the fusion module. That may be desirable or not, depending on the application. In the future, when we derive a satisfying story of how optimization algorithms interact with learning diagrams, one primary goal will be to understand how to describe mathematically models that are fixed vs allowed to vary.
>
> Your question appears to be more interested in dynamic routing, perhaps where a single expert is chosen given an input datum. It would seem we cannot accommodate truly discrete choices as we require that our maps be continuous. An extension to allow for some discontinuities (such as argmax) seems like an interesting direction for future work.  One idea we have for dynamic routing is to use the following [diagram](https://q.uiver.app/#q=WzAsMyxbMCwwLCJYIl0sWzIsMCwiWF5OIl0sWzQsMCwiWV5OIl0sWzAsMSwicm91dGVyIiwxXSxbMSwyLCJmX24iLDEseyJjdXJ2ZSI6M31dLFsxLDIsImZfMSIsMSx7ImN1cnZlIjotM31dLFsxLDIsImZfaSIsMSx7ImN1cnZlIjotMX1dLFsxLDIsImZfaiIsMSx7ImN1cnZlIjoxfV1d)
>
> In this diagram we design a router that we envision acting somewhat like an attention mechanism. The router injects injects a data point $x$ into a tuple ($a_1$ * $x$, $a_2$ * $x$, ..., $a_N$ * $x$) where the constants $a_i$ vary the strength of the input data assigned to expert i. Training the router means learning which expert to weight most strongly, i.e. "select", given an input datum.

---

### Official Review · Reviewer_u1cH · 2024-11-05

**Soundness:** 3
**Presentation:** 3
**Contribution:** 2
**Rating:** 3
**Confidence:** 5

**Summary:**

This work formalizes machine learning training using learning diagrams.
A learning diagram is defined as a directed graph that allows the composition of learning components, models, and loss functions.
The work uses a colimit construction for building training pipelines as diagrams which consist of sub-diagrams. Finally, a softwware library, DiagrammaticLearning.jl, implements the provided formulation.

**Strengths:**

1. This work presents an elegant mathematical notation and formulation of automated machine learning (AutoML) based on directed graphs consisting of machine learning primitives and losses. The work formalizes the creation of machine learning pipelines.

2. A software library is presented that implements this formulation.

**Weaknesses:**

1. It is unclear what advantage this framework provides over previous work on automated machine learning, specifically AutoML systems based on directed graphs of machine learning components, or machine learning pipelines.

2. The empirical performance improvement over trivial baselines, as described in Tables 2 and 3, is not significant: 1-3% compared to a weak baseline.

3. The work lacks novelty with respect to the broad field of AutoML, which consists of machine learning primitives, automated machine learning systems, and user interfaces, developed over the past decade:

a. See for example derivatives of the D3M project.

b. A book on AutoML: Automated Machine Learning, Hutter, Kotthoff, and Vanschoren, Springer Nature, 2019

c. AutoML systems:
Auto-sklearn, Google Cloud AutoML, H2O AutoML, AlphaD3M, Auto-Keras, AutoGluon, to name a few,

d. Ignoring an entire field of work presented in AutoML conferences and workshops since 2016.

**Questions:**

What capability does this formalism provide that is unavailable in existing AutoML systems?

---

> ### Author Response · Authors · 2024-11-23
>
> We would like to begin by thanking the reviewer for bringing our attention to the term "AutoML". While we were familiar with techniques from sub-fields such as neural architecture and hyperparameter search, we feel that literature in the AutoML space will greatly bolster our related work section. After review, we feel the AutoML literature does not significantly affect the significance or novelty of our contributions. Please see the updated section 5 and 6 for new references and our comparisons to them.
>
> > The empirical performance improvement over trivial baselines, as described in Tables 2 and 3, is not significant: 1-3\% compared to a weak baseline
>
> We refer the reviewer to Section 2 where we state "Our goal in this section is to show that our approach to developing training loops is expedient and feature rich without negatively affecting the final performance of the produced models". We have included language that emphasizes we are not trying to outperform existing baselines, just to show that our framework can replicate results with functionality that makes it convenient to do so. Please let us know if this is still unclear.
>
> > What capability does this formalism provide that is unavailable in existing AutoML systems?
>
> For one, no formalism we can find automatically builds a composite loss that trains multiple models to agree with one another. We welcome suggestions indicating how we can be more clear that this is a novel contribution. We believe that the number of ML training paradigms that can be framed as commuting diagrams shows the broad applicability of training models to make paths commute. Secondly, existing approaches focus on pipelines which might be capable of solving many kinds of tasks (e.g [1]), but focus on solving one given task at a time. The only work we can find that can juggle multiple tasks at a time is [2], which focuses on multi-task bayesian optimization and limits to settings with a single modality of data for which there may be multiple tasks. In contrast, our approach allows for arbitrarily many domains and modalities of data that may or may not be related by complicated chains of models. Thirdly, composition of models via universal constructions, as illustrated in Figure 2 of our manuscript, is not possible in any framework we have seen. We believe that this feature is quite convenient for building large diagrams. Fourthly, we can precisely relate the data of distinct machine learning training regimes via the category theoretic notion of a morphism. Doing so could allow for provenance of ML models to be captured as foundations are expanded upon, as well as for pattern matching in model zoos.
>
> These functionalities are made possible because we have a mathematical formalism that goes well beyond the typical refrain of "ML pipelines are DAGs". We believe the pursuit of formality is not simply academic but rather a critical step on the way to improved building, sharing, and reproduction of machine learning models. One clear advantage of our formalism over prior work that we've found is that it exists in a mathematical plane and is therefore "implementation free" in the sense that any library which faithfully reproduces the math will yield equivalent results. Rather than adhering to the view that "the reproducibility of ML models can be achieved by the preservation of the ML methods and models, the version of the programming language, the versions, and the names of the external libraries. Also, the image of the virtual machine or a whole file system can be created" [3] we believe reproducibility is achievable at a higher level of abstraction.
>
> Comparison with NAS algorithms is somewhat out of scope, but we'll conclude by noting that most work we've found on NAS does not formally define the operations that constitute a search space aside from saying they belong to a set, and even those that e.g. prescribe grammars [4] to structure the way in which composite systems can be composed do so in a somewhat ad-hoc way. Of course we may have missed literature which explicitly states that neural networks are DAGs, what kinds of spaces neural networks operate on (not specific data sets!), and what kinds (not sets of pytorch functions!) of operations between spaces are allowed. ***Defining this data is, modulo laws on composition and identities, the same process as defining the data of a category theoretic diagram.*** Therefore, if such references exist, we are providing a contribution by explicitly identifying the formal mathematics that underpin their techniques.
>
> [1] Lopez, Roque, et al. "AlphaD3M: An Open-Source AutoML Library for Multiple ML Tasks" 2023
>
> [2] Klein et al. “RoBO: A Flexible and Robust Bayesian Optimization Framework in Python” 2017
>
> [3] Nikitin et al. “Automated evolutionary approach for the design of composite machine learning pipelines” 2022
>
> [4] Drori et al. “Automatic Machine Learning by Pipeline Synthesis using Model-Based Reinforcement Learning and a Grammar” 2019

---

### Author Response · Authors · 2024-11-23
**General Response to Reviewers**

We would like to thank the reviewers for their very helpful feedback. Their questions and suggestions have allowed us to clear up our presentation of the material, improving the quality overall.

We address specific concerns, but as a general overview, this feedback allowed us to
- Strengthen our references with sources from the field of AutoML. While we find that our methods are significantly different and novel compared to many of these methods, having more comparison points helps our work stand out.
- Apply our methods to a complex, non-standard workflow, allowing us to point out further strengths of our generality while finding further extensions to our work for the future.
- Clear up the example demonstrating our modeling language. We rewrite this example in Julia rather than Python, as the language features allow for more readable code.

We again express our gratitude at this feedback, which has allowed us to present our work in a clearer way.

---

### Meta-Review · Area_Chair_VqDB · 2024-12-23

**Metareview:**

Summary
=======
The paper introduces "learning diagrams," a formal framework based on category theory for describing machine learning training protocols. Learning diagrams are directed graphs where nodes represent data spaces, intermediate representations, or parameter spaces (each with associated loss functions), and edges represent transformations between spaces. The framework enables composition of learning components through parallel paths, with training accomplished via a composite loss function. The authors implement this framework in a software library called DiagrammaticLearning.jl and demonstrate its application to various training paradigms including image captioning, knowledge distillation, and few-shot learning.

Strengths
=======
* Novel theoretical contribution applying category theory to formalize various ML training setups
* Represents training frameworks as data structures rather than code, increasing transparency
* Clear mathematical foundation for composing ML components
* Comprehensive demonstration across multiple training paradigms
* Well-structured presentation with increasingly complex examples
* Potential to improve reproducibility and maintainability of ML systems
* The introduction of DiagrammaticLearning.jl provides a practical tool for implementing the proposed framework.

Weaknesses
==========
* The work appears to overlap with existing AutoML research and tools
* Limited empirical evidence of advantages over conventional methods (only 1-3% improvement over baselines)
* Insufficient comparison to existing AutoML systems and frameworks and unclear practical advantages over them
* Implementation details are sparse
* Code representation is not intuitive despite the graphical framework
* No publicly available code in supplementary materials for reviewer evaluation

Reasons for decision
================
The paper presents an original theoretical framework with potential significance for ML system design and reproducibility. The implementation and empirical results are somewhat lacking.  The marginal performance improvements and lack of substantial novelty within the established AutoML landscape undermine the paper's impact. Additionally, concerns regarding usability and the absence of accessible code hinder the practical adoption of the proposed framework.

**Additional Comments On Reviewer Discussion:**

The most negative comments were by Reviwer u1cH (rating: Reject), concerning the advantage over the current rich solutions offered by the AutoML framework. The authors argued strongly for their case. In balance, the formulation is useful, but the practical demonstrations seem not yet convincing for practitioners.

---

### Decision · Program_Chairs · 2025-01-22

Accept (Poster)